# Lifestyle Risk Behaviours and Nutritional Status Associated with Mental Health Problems among Myanmar Adolescents: Secondary Analysis of a Nationwide 2016 School Survey

**DOI:** 10.3390/ijerph20176660

**Published:** 2023-08-27

**Authors:** Tin Zar Win, Yasuhiko Kamiya, Chris Fook Sheng Ng, Chris Smith, Su Myat Han

**Affiliations:** 1School of Tropical Medicine and Global Health, Nagasaki University, Nagasaki 852-8523, Japan; tinzarwin1992.uch@gmail.com (T.Z.W.); ykami@nagasaki-u.ac.jp (Y.K.); chrisng@m.u-tokyo.ac.jp (C.F.S.N.); pearl.june@gmail.com (S.M.H.); 2Department of Global Health Policy, School of International Health, Graduate School of Medicine, The University of Tokyo, Tokyo 113-8654, Japan; 3Department of Clinical Research, Faculty of Infectious and Tropical Diseases, London School of Hygiene & Tropical Medicine, London WC1E 7HT, UK; 4Department of Infectious Disease Epidemiology, Faculty of Epidemiology and Population Health, London School of Hygiene and Tropical Medicine, London WC1E 7HT, UK

**Keywords:** lifestyle behaviours, mental health, adolescents

## Abstract

Engaging in unhealthy lifestyles may be considered a risk factor for mental health problems, but there is limited evidence. This study aimed to identify the relationship between unhealthy lifestyles and mental health problems among Myanmar school-going adolescents. Global School Based Student Health Survey (GSHS) data from 2838 school-going adolescents from Myanmar were analysed. Bivariable and multivariable logistic regression analyses were applied. After adjusting for confounding variables, adolescents who were seated for more than three hours per day had higher odds of loneliness, anxiety-induced sleep disturbance, suicide ideation, and suicide attempts compared to others. Moreover, students who ate fruit less than one time per day were more likely to experience anxiety-induced sleep disturbance and suicidal ideation. Being a current drinker was significantly associated with suicidal ideation and attempt. Obese students were more likely to feel lonely compared to normal weight students. Our study indicates there is a strong association between unhealthy lifestyle behaviours and mental health problems among school adolescents in Myanmar.

## 1. Introduction

Mental health problems are increasing [1], making a significant contribution to the worldwide burden of disease [2]. Simultaneously, the number of adolescents reporting poor mental health is increasing. It is estimated that mental health problems affect 10–20% of adolescents worldwide. In 2019, one in seven youths aged 11–19 suffered from mental health conditions. Suicide is the third leading cause of death among adolescents aged 15–19 years, and more than 90% of suicides among adolescents occur in low- and middle-income countries [3]. Given these circumstances, the WHO stated mental health issues as a key priority for public health programs [4].

As stated in the biopsychosocial model, psychological factors (such as thoughts, emotions, and behaviours) and social factors (such as family, community, culture, and socioeconomic status) play roles in mental health [5]. Previous studies also reported that lifestyles factors and nutritional status are significantly associated with both physical and mental health [6,7]. Adolescence is a crucial stage for the development of lifelong behaviours [8]. Once unhealthy lifestyle behaviours are adopted, these often manifest as health problems in adulthood. The WHO has specifically defined adolescents as individuals between the ages of 10 and 19 [9]. In Myanmar, this demographic represents 19.4% of the population [10], making the matter particularly pressing in that country. The 2007 Myanmar GSHS reported that 2.2% of students experienced anxiety-induced sleep disturbance, 4.2% experienced loneliness, and 1.1% considered suicide in the past 12 months [11]. Poor mental health may influence adolescents’ quality of life, educational achievements, physical health, and social interactions. Long-lasting effects may impact their future employment, income potential, and overall well-being [12]. Adolescent mental health was recognized as an important focus for the Myanmar Five-year National Strategic Plan for Young People’s Health (2016–2020) [13]. However, very little is known about the factors associated with mental health conditions among adolescents, particularly the influence of lifestyle behaviours and nutritional status [14].

To address this gap, our study explores the association between unhealthy lifestyle behaviours, social factors, and mental health problems among adolescents in Myanmar using the second Global School-Based Student Health Survey (GSHS) 2016. The findings from the study aim to inform policy decision makers in designing and implementing effective programs to improve the mental well-being of school children.

## 2. Materials and Methods

### 2.1. Data Source and Procedure

We used data from the Global School-Based Student Health Survey (GSHS) 2016, Myanmar, to assess lifestyle risk behaviours, nutritional status, and their association with mental health problems among Myanmar adolescents. The GSHS is a cross-sectional survey conducted among school adolescents, developed by the World Health Organization with the assistance of the Centre for Disease Control and Prevention (CDC). The survey has a two-stage cluster sampling design. In the first stage, 50 schools were randomly chosen with a probability proportional to enrolment size. In the second stage, classes within the selected schools were randomly selected, and all students in selected classes were eligible to participate. Data from the survey consisted of 2838 students in grade 7–10 from 45 schools collected nationally. The overall response rate was 86.0%. Details of sampling methods and questionnaires are described in the GSHS 2016 report [10].

### 2.2. Measures

The details of the baseline characteristics of school adolescents are shown in Table 1. The study variables are detailed in Appendix A Table A1.

### 2.3. Data Management and Statistical Analysis

To obtain estimates that accurately reflect the entire population, we applied weighting to the data to account for differences in selection probabilities and school sizes within the two-stage sampling design. Additionally, weighting was performed to account for non-response, ensuring the inclusion of students who were selected but did not provide data. Specifically, we used the “svy” estimation command in STATA to incorporate the weight, stratum, and primary sampling unit (PSU) information provided in the dataset [15,16]. The weight used for the estimation was calculated as W = W1 × W2 × f1 × f2 × f3, where W1 was the inverse of the probability of selecting the school, W2 was the inverse of the probability of selecting the classroom within the school, f1 was a school-level non-response adjustment factor calculated by school size, f2 was a school-level non-response adjustment factor calculated by class, and f3 was a post stratification adjustment factor calculated by grade [10]. Descriptive analyses were conducted for all categorical variables using frequency and percentages. For bivariable analyses, we employed the Pearson chi-square test and logistic regression to investigate the associations between the outcome variables (mental health problems) and each independent variable. Independent variables displaying significant association with an outcome (*p* < 0.05) were considered possible confounders and included in the multivariable logistic regression model. Sociodemographic variables such as age, sex, and food insecurity were treated as a priori confounders and included in the model regardless of the significance of their associations. We reported odds ratios (OR) and adjusted odds ratios (AOR) along with their corresponding 95% confidence intervals and *p*-values, with *p* < 0.05 indicating statistical significance. All the data analyses were carried out in STATA package 15.1.

## 3. Results

### 3.1. Baseline Characteristics

A total of 2838 Myanmar students in grades 7–10 completed the self-reported questionnaire. The sociodemographic characteristics of participants are presented in Table 1. Among the students, 57.3% were under the age of 14, 53.0% were female, and 2.3% experienced food insecurity, indicating they were mostly or always hungry.

Approximately one-third of the participants were not physically active for at least one day during the past seven days. Additionally, 15.9% engaged in sedentary activities for more than three hours per day. Less than half of the students reported consuming fruit less than once per day, while 13.4% consumed vegetables less than once per day. Less than half of the participants consumed soft drinks once or more per day, and 4.5% were current drinkers. The proportions of students who were underweight, overweight, and obese were 15.3%, 5.1%, and 1.7%, respectively.

Over half of the participants (52.5%) reported that their parents understood their problems and were worried most of the time or always during the past 30 days. Similarly, 56.9% of participants had parents who knew what they were doing with their free time most of the time or always, and 46.8% of participants had parents who checked most of the time or always to see if their homework was done. The majority of the participants (78.0%) reported that their parents never, rarely, or sometimes went through their things without their approval.

Only a few adolescents (3.8%) reported not having any close friends, while over one-third of the adolescents reported having friends who were kind and helpful most of the time or always. Approximately 45.6% of the adolescents reported being bullied one day or more during the past 30 days.

Of the students, 8.6% felt lonely most of the time or always, 8.6% considered attempting suicide during the past 12 months, and 8.5% attempted suicide one or more times during the past 12 months. Furthermore, 3.7% of the adolescents experienced sleep disturbance due to worry, and 6.5% of the adolescents had attempted suicide during the past 12 months.

### 3.2. Lifestyle Risk Behaviours, Social Factors, and Loneliness

The logistic regression analysis showed that loneliness was associated with sedentary activity (OR 2.69, 95% CI 1.94, 3.72; *p* < 0.05), alcohol drinking (OR 2.69, 95% CI 1.94, 3.72), underweight (OR 0.57, 95% CI 0.41, 0.80), and obesity (OR 0.57, 95% CI 0.41, 0.80) (Table 2). In the multivariable analysis, adolescents engaging in sedentary activities for more than three hours per day had 2.46 times (95% CI 1.73, 3.50) higher odds of experiencing loneliness compared to those without sedentary behaviour. Additionally, students who were obese were more likely to feel lonely compared to students with a normal weight (OR 1.89, 95% CI 1.02, 3.48).

### 3.3. Lifestyle Risk Behaviours, Social Factors, and Worry-Induced Sleep Disturbance

There was a significant association between sedentary activities, fruit intake, and current drinking habits and worry-induced sleep disturbance (Table 3). After adjusting for age, sex, food insecurity, sedentary behaviour, fruit intake, current drinker, no close friend, and been bullied, adolescents who had experienced being bullied were more than four times (95% CI 2.52, 6.65) more likely to have anxiety-induced sleep disturbances compared to those without. The adolescents who spent more than 3 h per day engaged in sedentary activities had 2.32 times (95% CI 1.61, 3.34) higher odds of experiencing anxiety-induced sleep disturbances compared to those without sedentary behaviour. Students who consumed fruit less than once per day were more likely to experience anxiety-induced sleep disturbance (AOR 1.82, 95% CI 1.25, 2.64) (Table 3).

### 3.4. Lifestyle Risk Behaviours and Suicide Ideation

In the bivariable analysis, suicidal ideation was significantly associated with sedentary behaviour (OR 2.25, 95% CI 1.67, 3.05), inadequate intake of fruit (OR 1.54, 95% CI 1.19, 2.01), and being a current alcohol drinker (OR 3.79, 95% CI 2.63, 5.46) (Table 4). The associations remained in the multivariable analysis. Adolescents with sedentary behaviours were 1.85 times (95% CI, 1.24, 2.77) more likely to have suicide ideation compared to adolescents who sat for less than 3 h per day. Students who ate fruit less than one time per day had 1.44 times (95% CI 1.05, 1.98) higher odds of suicide ideation. Currently drinking alcohol was associated with suicidal ideation (AOR 3.29, 95% CI 1.83, 5.91).

### 3.5. Lifestyle Risk Behaviours and Suicide Attempt

In the bivariable analysis, previous suicide attempts were associated with adolescents experiencing food insecurity (OR 2.72, 95% CI 1.68, 4.43), current drinkers (OR 4.27, 95% CI 2.84, 6.40), and individuals with a sedentary lifestyle (OR 2.12, 95% CI 1.51, 2.98) (Table 5). In the multivariable analysis, sedentary behaviour remained significantly associated with suicide attempts (AOR 1.76, 95% CI 1.17, 2.66). Moreover, students who consumed at least one alcoholic drink in the past 30 days had a higher risk of attempting suicide (AOR 3.75, 95% CI 1.80, 7.81). Conversely, underweight students had a lower risk of suicide attempts compared to students with a normal weight (AOR 0.38, 95% CI 0.20, 0.72).

## 4. Discussion

Mental health problems among adolescents are a global concern, and understanding the modifiable risk factors associated with these issues is crucial for policymakers, teachers, and communities. The present study examined the association between lifestyle risk behaviours, social factors, and mental health problems among school adolescents in Myanmar. The main findings indicate that high sedentary behaviour is positively linked to feelings of loneliness, anxiety-induced sleep disturbance, suicidal ideation, and suicide attempts. Consuming fruit less than once a day is associated with increased odds of worry-induced sleep disturbance and suicidal ideation. Drinking alcohol at least once in the past 30 days is associated with suicidal ideation and suicide attempts. Obesity is significantly associated with feelings of loneliness, while being underweight is negatively associated with suicide attempts.

### 4.1. Mental Health Problems

The prevalence of mental health problems among adolescents in Myanmar has increased considerably. GSHS data from 2007 in Myanmar reported that the percentage of students seriously considering suicide was 1.1% [11]. In the current study, the prevalence of suicidal ideation was 8.6%. Popularity in smartphone and social media use may impact adolescents’ self-view through social comparison, which might promote self-harm and suicidal thoughts [17]. Female students were found to be more susceptible to mental health problems. Gender-based violence and negative life experiences may contribute to the higher prevalence of mental health issues among females [18]. However, the percentage of mental health problems was still lower compared to other countries, for example, 14.0% in Nepal [19], and 17.2% in the U.S. [20]. The extended family structure and the influence of Buddhism in Myanmar may contribute to the differences in the prevalence of mental health problems compared to other countries. The extended family structure may have a positive impact on children’s self-esteem, which is a fundamental aspect of mental health [21]. Additionally, meditation, a Buddhist practice, can help reduce stress, anxiety, and depression [22].

### 4.2. Risk Factors: Physical Activity and Sedentary Behaviours

The current study did not find a significant association between physical inactivity and mental health problems such as loneliness, anxiety, and suicidal risk, contrary to previous evidence from 52 low- and middle-income countries [23]. The WHO recommends that adolescents engage in at least 60 min of moderate-to-vigorous intensity physical activity per day. Around 31% of adolescents in this study did not meet this recommendation, likely due to a lack of physical facilities in most schools in Myanmar. The insufficient knowledge about the positive effects of physical activity and the popularity of motorbike usage among adolescents should also be considered. Previous research has shown that participation in sports is more important than overall physical activity when it comes to suicidality [24]. However, the Myanmar culture does not encourage girls to participate in sports. Women running and playing are viewed as against the norms of a woman’s gentleness, grace, and modesty. Future studies should consider sports participation as an important factor for mental health. Moreover, the burden of a sedentary lifestyle in Myanmar has been increasing compared to 2007 [25]. This upward trend can be attributed to advancements in technology and globalization. Our findings demonstrate that sedentary behaviours among adolescents are associated with mental health outcomes, such as loneliness, anxiety-induced sleep disturbance, suicidal ideation, suicidal planning, and suicide attempts, and are consistent with previous research that has reported a significant association between sedentary behaviour and mental health problems among adolescents in the United States [20,26], as well as in 52 low- and middle-income countries [23]. One possible explanation for this finding is that high levels of sedentary behaviour can lead individuals to withdraw from interpersonal relationships, potentially resulting in mental health problems. Sedentary behaviours encompass various activities such as TV watching, video game playing, computer use, and reading. A meta-analysis study reported that the effect of sedentary behaviour on mental health depends on the specific type of sedentary behaviours [27]. In today’s world, children and adolescents extensively use technology and screens. Spending more than 2–3 h per day in front of screens may lead to developmental delays in language acquisition and communication skills in children. Furthermore, excessive screen time is associated with both sleep quality and duration. All these circumstances contribute to the worsening of mental health conditions such as depression, anxiety, and low self-esteem, and increasing vulnerability to suicidal tendencies.

Our study found that 41% of participants reported consuming fruits less than once a day, and 13% indicated consuming vegetables less than once a day. A systematic review has shown that parent factors, such as education and occupation, are positively associated with fruit and vegetable consumption among adolescents [28], which is also likely applicable to Myanmar. Due to the limitation of the survey, we were not able to assess the education and occupation of the parents of the participants in this study. Our findings regarding the association between dietary habits and mental health problems align with some studies but differ from others conducted among sub-Saharan African adolescents [29]. The differences in findings may be attributable to the type of survey. The questions regarding fruit and vegetable consumption in the Global School-based Student Health Survey (GSHS) asked about experiences in the past 30 days, which may introduce recall bias. Recall bias may lead to an overestimation or underestimation of the frequency of consumption and may impact the strength of the associations.

The findings of the current study revealed that approximately 43.0% of students consumed soft drinks one or more times per day. Sugar stimulates dopamine production in the brain, activating pleasure centres [30]. Soft drinks contain high amounts of sugar, which may contribute to adolescents developing an addiction to sugary beverages. Consistent with a previous study conducted among five ASEAN states [31], our study did not find any significant association between soft drink intake and mental distress. The positive association between soft drink intake and mental health is more statistically significant in middle-income countries compared to high-income or low-income countries [32]. The difference may be the amount and type of soft drinks consumed between these countries. The GSHS questionnaire did not include a question related to the amount of soft drink intake, which could be a potential reason for our findings. The study warrants further studies to include the amount and type of soft drinks (e.g., energy drinks) consumed in the survey. 

In comparison to other countries, such as Thailand with a 23.0% prevalence of current drinkers and Bhutan with a 24.2% prevalence [19], the prevalence of current drinkers among adolescents in Myanmar is lower (4.5%). The impact of religion and culture should be considered as a possible reason for this variation. Similar to previous research [33,34,35], the current study found significant associations between alcohol consumption and suicidal risks. Alcohol misuse can lead to suicidal thoughts through disinhibition, impulsiveness, and impaired judgment. A previous multi-country study concluded that alcohol use affected the mental health of adolescents [36], but contrary to that study, our study did not find a significant association between current drinkers and feelings of loneliness or anxiety-induced sleep disturbance. One possible reason may be the different forms of alcohol consumption. In Myanmar, people rarely drink alone and prefer to enjoy drinking with friends, which may even alleviate feelings of anxiety. Additionally, given the low level of mental health awareness in Myanmar, individuals may not recognize their own experiences of loneliness and anxiety, potentially impacting their responses to the GSHS questionnaire.

The prevalence of overweight/obesity was higher compared to 2007 GSHS data in Myanmar [11]. This may be due to the influence of western culture on diet habits through social media. Studies investigating the relationship between body weight and its impact on mental health, such as depression and suicidal risks, demonstrated that those who have a high level of BMI or are underweight are more likely to have increased an risk of depression and suicidal attempts. However, our study shows that underweight adolescents have a lower risk of mental health problems compared to normal-weight adolescents. Furthermore, the study could not find any significant association between overweight and suicidal vulnerability. It is not clear why underweight students are less likely to attempt suicide compared to normal-weight students. These days, young people are obsessed with having a slim body, so it is possible that they do not perceive themselves as underweight, and may be satisfied with their body. Previous studies among Korean adolescents [37] and US adolescents [38] indicated that body weight perception is an important risk factor for suicidal ideation compared to BMI. The perception of body weight may affect mental health more than the actual body weight.

## 5. Limitations and Implications for Future Research

This study had some limitations. Firstly, there was a lack of key information regarding the data, such as detailed information about participants and variable definitions. Secondly, it was a cross-sectional study, so it could not show a cause–effect relationship. Thirdly, the participants were only adolescents who were enrolled in school; therefore, the findings could not be generalized to all the adolescents in the country. It may be that adolescents outside school adopt unhealthy behaviours compared to school-going children. We recommend further longitudinal studies to explore the causal relationship between lifestyle risk behaviours and mental health issues among adolescents in and out of school.

Last but not least, the role of families in suicidality among adolescents has not been discussed as it is not the main research question. However, the study has shown that parents’ care was a protective factor for suicidal behaviour [39]. We recommend future research consider family data as crucial determinants in preventing adolescent mental health.

## 6. Conclusions

This study aimed to identify the impact of lifestyle risk behaviours on mental health among adolescents in Myanmar, specifically in 2016. It can be concluded that unhealthy behaviours such as sedentary behaviour, inadequate consumption of fruits and vegetables, high intake of soft drinks, alcohol consumption, and body weight were significantly associated with feelings of loneliness, anxiety-induced sleep disturbance, and suicidal risks among adolescents in Myanmar. These findings underscore the need for implementing programmes that promote and prevent mental health issues by addressing the identified lifestyle risk behaviours among adolescents in Myanmar. Further longitudinal studies are recommended to explore the causal relationships and assess the impact of the quantity and type of dietary habits, as well as specific sedentary activities, on mental health.

## Figures and Tables

**Table 1 ijerph-20-06660-t001:** Baseline characteristics (unweighted) among Myanmar school adolescents, 2016 GSHS.

Characteristics	Number (Percentage)
Age group	
≤14	1626 (57.3)
≥15	1202 (42.4)
Sex	
Male	1301 (45.8)
Female	1511 (53.2)
Body Mass Index (BMI)	
Normal weight	1922 (67.7)
Underweight	433 (15.3)
Overweight	146 (5.1)
Obese	49 (1.7)
Lifestyle behaviours factors	
Food insecurity	66 (2.3)
Physical inactivity	878 (30.9)
Sitting activities more than 3 h per day	450 (15.9)
Ate fruit less than one time per day	1165 (41.0)
Ate vegetable less than one time per day	381 (13.4)
Intake of soft drinks one or more times per day	1212 (42.7)
Current drinker	127 (4.5)
Social Factors	
Parental understanding	1491 (52.5)
Parental monitoring	1616 (56.9)
Parental check	1327 (46.8)
Parental privacy allowance	2213 (78.0)
Having no close friends	108 (3.8)
Having supportive friends	1047 (36.9)
Experience of being bullied	1295 (45.6)
Mental Health status	
Loneliness	245 (8.6)
Worried induced sleep disturbance	104 (3.7)
Suicidal ideation	245 (8.6)
Attempted suicide	240 (8.5)

**Table 2 ijerph-20-06660-t002:** Association between lifestyle risk behaviours and loneliness among Myanmar school adolescents, 2016 GSHS.

Characteristics	Loneliness			
	OR (95% CI)	*p*-Value	AOR * (95% CI)	*p*-Value
Age group				
≤14	1	0.003	1	0.008
≥15	1.55 (1.19, 2.03)		1.50 (1.12, 2.01)	
Sex				
Female	1	0.064	1	0.029
Male	0.68 (0.46, 1.02)		0.61 (0.40, 0.95)	
Nutritional status				
Normal weight	1		1	
Underweight	0.57 (0.41, 0.80)	0.002	0.65 (0.43, 1.00)	0.051
Overweight	0.89 (0.41, 1.94)	0.756	0.96 (0.39, 2.36)	0.920
Obese	2.26 (1.09, 4.72)	0.031	1.89 (1.02, 3.48)	0.042
Lifestyle behaviours factors				
Food insecurity	2.70 (1.26, 5.76)	0.013	1.47 (0.57, 3.81)	0.408
Physical inactivity	1.34 (1.00, 1.79)	0.052	1.33 (0.96, 1.83)	0.080
Sitting more than 3 h per day	2.69 (1.94, 3.72)	<0.001	2.46 (1.73, 3.50)	<0.001
Fruit intake less than one time per day	1.37 (0.99, 1.91)	0.060	n/a	
Vegetable intake less than one time per day	1.15 (0.81, 1.64)	0.407	n/a	
Soft drinks intake one or more times per day	1.17 (0.88, 1.56)	0.259	n/a	
Current drinker	1.94 (1.08, 3.48)	0.028	1.34 (0.67, 2.66)	0.390
Social Factors				
Parental understanding	0.61 (0.44, 0.86)	0.006	0.75 (0.51, 1.10)	0.136
Parental monitoring	0.64 (0.49, 0.82)	0.001	0.70 (0.52, 0.94)	0.019
Parental check	0.68(0.49, 0.95)	0.024	0.84(0.58, 1.22)	0.352
Parental privacy allowance	0.70 (0.51, 0.95)	0.027	0.51 (0.35, 0.76)	0.002
Having no close friends	3.40 (1.93, 5.99)	<0.001	2.83 (1.47, 5.45)	0.003
Having supportive friends	1.16 (0.90, 1.50)	0.239	n/a	
Experience of being bullied	3.30 (2.39, 4.55)	<0.001	3.08 (2.10, 4.52)	<0.001

* Adjusted for age, sex, food insecurity, physical inactivity, sedentary behaviour, current drinker, nutritional status, parent understanding, parent monitoring, parent check, parent allow privacy, no close friend, and been bullied. GSHS—global school-based student health survey; cut-off—0.05; OR—odd ratio; AOR—adjusted odd ratio.

**Table 3 ijerph-20-06660-t003:** Association between lifestyle risk behaviours and worry-induced sleep disturbance among Myanmar school adolescents, 2016 GSHS.

Characteristics	Worry-Induced Sleep Disturbance			
	OR (95% CI)	*p*-Value	AOR * (95% CI)	*p*-Value
Age				
≤14	1		1	
≥15	1.30 (0.78, 2.15)	0.297	1.12 (0.74, 1.70)	0.587
Sex				
Female	1		1	
Male	0.88 (0.62, 1.23)	0.431	0.68 (0.49, 0.94)	0.023
Body Mass Index (BMI)				
Normal weight	1		n/a	
Underweight	0.51 (0.22, 1.21)	0.119		
Overweight	1.51 (0.54, 4.20)	0.413		
Obese	1.99 (0.34, 11.84)	0.432		
Lifestyle behaviours factors				
Food insecurity	3.46 (1.46, 8.19)	0.007	2.39 (0.87, 6.56)	0.087
Physical inactivity	1.15 (0.69, 1.91)	0.574	n/a	
Sitting more than 3 h per day	2.52 (1.78, 3.57)	<0.001	2.32 (1.61, 3.34)	<0.001
Fruit intake less than one time per day	1.65 (1.15, 2.38)	0.009	1.82 (1.25, 2.64)	0.003
Vegetable intake less than one time per day	1.24 (0.68, 2.25)	0.473	n/a	
Soft drinks intake one or more times per day	1.15 (0.69, 1.92)	0.571	n/a	
Current drinker	2.74 (1.31, 5.70)	0.009	1.42 (0.66, 3.06)	0.353
Social Factors				
Parental understanding	0.69 (0.42, 1.13)	0.134		
Parental monitoring	0.78 (0.45, 1.38)	0.381		
Parental check	0.97 (0.65, 1.43)	0.864		
Parental privacy allowance	0.61 (0.36, 1.02)	0.058		
Having no close friends	3.76 (2.03, 6.99)	<0.001	3.23 (1.87, 5.57)	<0.001
Having supportive friends	0.79 (0.50, 1.25)	0.293		
Experience of being bullied	4.41 (2.57, 7.55)	<0.001	4.10 (2.52, 6.65)	<0.001

* Adjusted for age, sex, food insecurity, sedentary behaviour, fruit intake, current drinker, no close friend and been bullied. GSHS—global school-based student health survey; cut-off—0.05l; OR—odd ratio; AOR—adjusted odd ratio.

**Table 4 ijerph-20-06660-t004:** Association between lifestyle risk behaviours and suicide ideation among Myanmar school adolescents, 2016 GSHS.

Characteristics	Suicide Ideation			
	OR (95% CI)	*p*-Value	AOR * (95% CI)	*p*-Value
Age				
≤14	1		1	
≥15	1.12 (0.69, 1.80)	0.638	0.99 (0.61, 1.61)	0.966
Sex				
Female	1		1	
Male	0.73 (0.57, 0.95)	0.019	0.48 (0.36, 0.65)	<0.001
Body mass index (BMI)				
Normal weight	1			
Underweight	0.67 (0.43, 1.05)	0.075		
Overweight	0.55 (0.25, 1.20)	0.126		
Obese	1.34 (0.39, 4.59)	0.622		
Lifestyle behaviours factors				
Food insecurity	3.94 (1.85, 8.40)	0.001	2.08 (0.93, 4.68)	0.073
Physical inactivity	1.38 (1.00, 1.91)	0.053	1.27 (0.96, 1.68)	0.089
Sitting more than 3 h per day	2.25 (1.67, 3.05)	<0.001	1.85 (1.24, 2.77)	0.004
Fruit intake less than one time per day	1.54 (1.19, 2.01)	0.002	1.44 (1.05, 1.98)	0.026
Vegetable intake less than one time per day	1.41 (0.77, 2.57)	0.248	n/a	
Soft drinks intake one or more times per day	1.21 (0.82, 1.79)	0.329	n/a	
Current drinker	3.79 (2.63, 5.46)	<0.001	3.29 (1.83, 5.91)	<0.001
Social Factors				
Parental understanding	0.39 (0.28, 0.55)	<0.001	0.58 (0.42, 0.81)	0.003
Parental monitoring	0.41 (0.31, 0.54)	<0.001	0.59 (0.41, 0.83)	0.004
Parental check	0.53 (0.39, 0.72)	<0.001	0.79 (0.55, 1.14)	0.201
Parental privacy allowance	0.76 (0.55, 1.05)	0.095	n/a	
Having no close friends	3.03 (1.59, 5.80)	0.002	2.24 (1.19, 4.20)	0.015
Having supportive friends	0.99 (0.75, 1.31)	0.964	n/a	
Experience of being bullied	3.47 (2.59, 4.64)	<0.001	2.74 (2.05, 3.66)	<0.001

* Adjusted for age, sex, food insecurity, physical inactivity, sedentary behaviour, fruit intake, current drinker, parent understanding, parent monitoring, parent check, no close friend, and been bullied. GSHS —global school-based student health survey; cut-off—0.05; OR—odd ratio; AOR—adjusted odd ratio.

**Table 5 ijerph-20-06660-t005:** Association between lifestyle behaviours and suicide attempt among Myanmar school adolescents, 2016 GSHS.

Characteristics	Suicide Attempt			
	OR (95% CI)	*p*-Value	AOR * (95% CI)	*p*-Value
Age				
≤14	1		1	
≥15	1.26 (0.77, 2.06)	0.340	1.15 (0.74, 1.79)	0.522
Sex				
Female	1		1	
Male	0.64 (0.47, 0.87)	0.006	0.46 (0.32, 0.66)	<0.001
Body Mass Index (BMI)				
Normal weight	1		1	
Underweight	0.31 (0.17, 0.57)	0.001	0.38 (0.20, 0.72)	0.004
Overweight	1.00 (0.54, 1.87)	0.993	1.03 (0.56, 1.90)	0.910
Obese	1.33 (0.57, 3.08)	0.494	1.32 (0.54, 3.25)	0.524
Lifestyle behaviours factors				
Food insecurity	2.72 (1.68, 4.43)	<0.001	1.54 (0.75, 3.15)	0.227
Physical inactivity	1.28 (1.02, 1.62)	0.037	1.20 (0.94, 1.53)	0.140
Sitting more than 3 h per day	2.12 (1.51, 2.98)	<0.001	1.76 (1.17, 2.66)	0.009
Fruit intake less than one time per day	1.23 (0.91, 1.67)	0.175	n/a	
Vegetable intake less than one time per day	1.28 (0.81, 2.02)	0.268	n/a	
Soft drinks intake one or more times per day	1.39 (0.96, 2.01)	0.077	n/a	
Current drinker	4.27 (2.84, 6.40)	<0.001	3.75 (1.80, 7.81)	0.001
Social Factors				
Parental understanding	0.48 (0.36, 0.64)	<0.001	0.59 (0.41, 0.85)	0.007
Parental monitoring	0.42 (0.32, 0.56)	<0.001	0.51 (0.36, 0.71)	<0.001
Parental check	0.62 (0.46, 0.84)	0.004	0.95 (0.68, 1.33)	0.768
Parental privacy allowance	0.85 (0.58, 1.25)	0.394	n/a	
Having no close friends	1.72 (0.81, 3.65)	0.149	n/a	
Having supportive friends	0.89 (0.69, 1.15)	0.357	n/a	
Experience of being bullied	3.62 (2.64, 4.96)	<0.001	2.66 (1.96, 3.61)	<0.001

* Adjusted for age, sex, food insecurity, physical inactivity, sedentary behaviour, fruit intake, current drinker, parent understanding, parent monitoring, parent check, no close friend and been bullied. GSHS—global school-based student health survey; cut-off—0.05; OR—odd ratio; AOR—adjusted odd ratio.

## Data Availability

A publicly available dataset was analysed in this study. This data can be found here: “https://extranet.who.int/ncdsmicrodata/index.php/catalog/GSHS (accessed on 23 June 2020)”.

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
