# Peer review of "Lifestyle Risk Behaviours and Nutritional Status Associated with Mental Health Problems among Myanmar Adolescents: Secondary Analysis of a Nationwide 2016 School Survey"

_ijerph, 2023, doi:10.3390/ijerph20176660_

Round 1
Reviewer 1 Report
The authors present the findings based on the data from a sample of 2838 students. The findings show that several behaviors are related to mental health problems. However, this manuscript contains a few issues that should be addressed before it will be suitable for publication.
The literature review for this study appears to have several areas for improvement. It is not clear how the literature reviewed is relevant to the specific results being addressed in this study. Another area for improvement is a more clear structure to the review. Providing a clear and logical structure to the review, such as full literature reviews (including characteristics of Myanmar adolescents), existing theoretical frameworks, and purpose of study (including research questions), would improve the readability and understanding of the review.
The research design for this study uses national datasets, which has strengths and limitations that could be discussed more extensively with a rationale provided. One weakness is lack of key information regarding the data such as participant information, collection procedures, variable definitions/description, and variable weight.
Please add sections for limitations and implications for future research.
N/A
Reviewer 2 Report
Fisrt of all thank the authors for the effort and for choosing such and important issue in mental health and general health issues.
There are few minor problems that can be solve in the review:
1.- In introduction the reference number 7 must be improve: "American College of Obstetricians and Gynecologists (2017) ‘Obesity in Adolescents ACOG committee opinion’, Obstetrics & Gynecology, 130(4), p. e210." I'm sure that there are references related to obesity and psychological problems better than this reference.
2.- In material an methods 2.3 Data management and statistical analysis, the first sentence: "We used the weight, stratum and PSU provided in the dataset for the analysis", must be explained deeply, and the words must be written before putting them in initials.
3.- In results 3.1. Baseline characteristics (line 22): "The proportions of students who felt lonely most of the time or always, considered attempting suicide during the past 12 months, and attempted suicide one or more times during the past 12 months were nearly the same (8.6%, 8.6%, and 8.5%, respectively)". It should be explained differently because it can lead to confusion that the three percentages are about the same subjects, being percentages without relationship between them.
4.- In results 3.2. Lifestyle Risk Behaviours, Social Factors and Loneliness the reference of Table 1 should be Table 2.
5.- In discussion 4.1 Mental Health Problems the sentence: "In the current study, the prevalence of suicidal ideation was 8.6%". Must be explained the difference between de data in the study of 8.6% and the data in other studies in Myanmar en 2007 of 1.1%
6.- In discussion 4.1 Mental Health Problems the sentence: "Female students were found to be more susceptible to mental health problems. Gender-based violence, low income, gender inequality, and negative life experiences may contribute to the higher prevalence of mental health issues among females", must be referenced.
7.- I understand that the objective and hypothesis of the study where related to lifestyle risk behaviours and nutricional status, but having interesting data related to parent relationships and bulling I recommend to broaden your results and discusion with these variables, or at least propose future studies using these extra datas. Maybe you can get some inspiration with the article: Alvarez-Subiela, X.; Castellano-Tejedor, C.; Villar-Cabeza, F.; Vila-Grifoll, M.; Palao-Vidal, D. Family Factors Related to Suicidal Behavior in Adolescents. Int. J. Environ. Res. Public Health 2022, 19, 9892. https://doi.org/10.3390/ijerph19169892 .
8.- In the references you can find some problems of editions with the numbers of the references repited, some color differences or other format problems.
Round 2
Reviewer 1 Report
accept in current form
accept in current form